# Surface Heterogeneous Nucleation-Mediated Release of Beta-Carotene from Porous Silicon

**DOI:** 10.3390/nano10091659

**Published:** 2020-08-24

**Authors:** Chiara Piotto, Sidharam P. Pujari, Han Zuilhof, Paolo Bettotti

**Affiliations:** 1Nanoscience Laboratory, Department of Physics, University of Trento, 38123 Povo, Trento, Italy; chiara.piotto@gmail.com; 2Laboratory of Organic Chemistry, Wageningen University, Stippeneng 4, 6708 Wageningen, The Netherlands; sidharam.pujari@wur.nl (S.P.P.); han.zuilhof@wur.nl (H.Z.); 3Department of Chemical and Materials Engineering, Faculty of Engineering, King Abdulaziz University, Jeddah 21589, Saudi Arabia; 4School of Pharmaceutical Sciences and Technology, Tianjin University, 92 Weijin Road, Tianjin 300072, China

**Keywords:** porous silicon, heterogeneous nucleation, drug delivery, carotene, nanopores

## Abstract

We demonstrate that the release of a poorly soluble molecule from nanoporous carriers is a complex process that undergoes heterogeneous surface nucleation events even under significantly diluted release conditions, and that those events heavily affect the dynamics of release. Using beta-carotene and porous silicon as loaded molecule and carrier model, respectively, we show that the cargo easily nucleates at the pore surface during the release, forming micro- to macroscopic solid particles at the pores surface. These particles dissolve at a much slower pace, compared to the rate of dissolution of pure beta-carotene in the same solvent, and they negatively affect the reproducibility of the release experiments, possibly because their solubility depends on their size distribution. We propose to exploit this aspect to use release kinetics as a better alternative to the induction time method, and to thereby detect heterogenous nucleation during release experiments. In fact, release dynamics provide much higher sensitivity and reproducibility as they average over the entire sample surface instead of depending on statistical analysis over a small area to find clusters.

## 1. Introduction

Nanoporous materials posess several properties that render them of interest for a wide range of different uses. The possibilities to finely tune their physico-chemical properties, often by simple modification of their synthesis, make nanoporous materials extremely flexible platforms to realize advanced materials with well-controlled and tunable characteristics. Porous materials are highly investigated for applications ranging from catalysis [1,2,3,4], via energetics [5,6,7], to biology [8,9,10,11]. Often the specific properties are due to the way the nanostructuring affects the interactions between the porous scaffold and the substances it comes in to contact with. For example, surface nanostructuring heavily modifies the energy landscape to induce surface nucleation and, consequently, liquid–solid transitions at the interfaces [12].

Nanoporous materials are ideal candidates to develop effective drug delivery systems (DDS) with tailored properties and release characteristics. Their large surface area, well controlled pore size distribution, tunable chemistry, and top-down fabrication techniques are some of the parameters that render them of interest in both research and industrial applications [13,14,15,16,17,18,19,20,21]. Among the plethora of materials studied for these purposes, porous silica and porous silicon (PSi) are some of the most investigated inorganic carriers thanks to their many interesting properties, such as: chemical stability and inertness, well-known and controlled surface chemistry, large tunability of the porous architectures spanning from the “truly” nanometric range (e.g., MCM41) up to micron-sized pores (e.g., macroporous silicon and reverse opals) and, finally, their biocompatibility (they are both approved for medical uses).

Moreover, PSi has some advantages compared to other materials, the most important are: (i) PSi nanoparticles are fluorescent under UV-blue excitation, making these carriers natively fluorescent tags, and permitting to follow their fate with fluorescent analysis [22,23]; (ii) the surface chemistry of the as-prepared PSi provides broadly tunable surface chemistry (an important aspect since the use of inorganic carriers inherently compatible with nonpolar drugs is fundamental as a large amount of newly developed drugs are lipophilic [24]): as-etched silicon surfaces slowly turn from lipophilic into polar by means of a spontaneous oxidation process [25,26] and several strategies have been developed to functionalize PSi surface: from common silanization [27] and alkene photoreactions [28] to more complex approaches, such as the carbonization reaction [29,30]; (iii) PSi pores can be arranged into highly ordered arrays [31], enabling the synthesis of materials with highly tailored release dynamics, particularly when pores are in the mesoscale range (20–200 nm).

Mesoscopic length scales (from tens to few hundreds of nm) are hard to model as they are at the edge between the nanoworld and the bulk realm, showing properties markedly different from both regimes. In fact, while PSi carriers with micron-sized pores offer no advantages over bulk DDS formulation, neither in terms of loading capacity nor of release dynamics, truly nanoporous carriers are not suited to load large molecules and are prone to pore clogging and superficial crystallizations during drug loading (as recently described also for PSi [32]). To estimate the type of diffusion expected in our system, we calculate the Knudsen number using:(1)$$K_{n}=\frac{k_{B}T}{\sqrt{2}\pi \sigma^{2}P\cdot L\;}$$
where *T* denotes system temperature, *k_B_* Boltzmann constant, *P* system pressure, *σ* particle size, and *L* characteristic system length. Assuming 300 K, atmospheric pressure, BCAR size of 2 nm and a typical length of 100 nm, we obtain a *K_n_* > 0.02 that indicates the transitional regime between viscous and molecular diffusion.

When working with a hydrophobic drug the control over its solidification is of utmost importance, as the drug will be typically surrounded by an aqueous environment where it easily nucleates into solid and non-bioavailable forms. While some strategies were developed to optimize the loading of lipophilic compounds, [33,34,35,36,37,38] no studies report on how the release dynamics is affected by the complex inter-relations between the structure of the carrier, the amount of drug loaded and possible crystallizations triggered during the release.

It is well known that nucleation is mostly driven by energetic balances, as it reduces the free energy barrier required to form critical nuclei. As demonstrated by several works, in most cases, the key process triggering nucleation is HN, rather than homogeneous nucleation [39,40]. Although different models have been proposed to describe HN, a complete theory is still missing. Liu reported a kinetic model for 3D HN that delineates the competitive mechanisms between nucleation, growth and their inter-relations [41], while other authors focused on the role of line tension [42,43] and correlate this parameter with experimental data. Recently, multiple nucleation paths have been proposed to describe HN: some studies focus on protein crystallization, while others demonstrate similar effects also for smaller molecules [44,45,46,47]. All these models assume that nucleation happens because of large density fluctuations that bring about the formation of pre-nuclei able to stimulate the phase transition.

When dealing with nanoporous materials, molecules reside in constrained volumes and this enclosure in nanopores can in fact stimulate several new phenomena, such as: the formation of polymorphs with largely different properties [48,49,50]; surface interactions leading to local solute enrichment that increase the supersaturation and the probability of HN, either nearby to or away from the pore surface, depending on the type of interactions [51]; the creation of local environments due to pores surface roughness with markedly different energy landscapes for HN. Thus, a plethora of nucleation sites are available [52]. In general, the exact growth mechanism changes depending on actual system conditions [41,53,54].

Recent works recognized the importance of heterogeneous processes in DDS, both to increase the industrial process efficiency (through so-called process intensification technology) [55] as well as to unveil their role in drug loading and crystallization [56].

In this work we consider the specific case of a poorly soluble molecule loaded in a submicron porous matrix: we focus on how molecules out-diffuse from the carrier, and we demonstrate a peculiar and never reported effect of surface heterogeneous nucleation (HN) even under highly undersaturated release conditions. We aim to clarify this process in a fundamental way, and discuss the possible implications of this complex dynamics, using as example the case of nanoporous materials in DDS. We demonstrate that surface HN produces macroscopic aggregates on the surface of porous carriers during the release of the cargo even in diluted solutions, and that these aggregates dissolve at very slow rate. Due to the stochastic nature of HN (in terms of both size and frequency), it results in a large increase in the variance of the release profiles and, consequently, a reduced reproducibility. Thus, this work highlights the importance of controlling the release by avoiding conditions that favor the formation of agglomerates during the release. In turn, this fact limits the maximum amount of compound that can be loaded in the carrier to produce reproducible release profiles. In addition, we propose the use of the release kinetics as a precise method to characterize the appearance of HN and as a substitute to the standard induction time method.

The study has been done on a model system, to highlight the role of HN on the release of poorly soluble molecules from PSi carriers even under largely undersaturated release conditions, yet similar effects are foreseen even if more soluble compounds are considered. We choose PSi as carrier for the reasons highlighted above and β-carotene (BCAR), as hydrophobic drug, as it owns: (a) a highly apolar structure (nominally BCAR has null electric dipole), (b) low water solubility (0.6 mg/L at 25 °C), and (c) a large optical absorbance (*ε* = 1.40 × 10^5^ cm^−1^ M^−1^ at 450 nm) that allows its quantification down to nmol concentrations.

## 2. Materials and Methods

### 2.1. Chemicals and Reagents

Silicon (100) wafers were supplied by University Wafers (South Boston, MA, USA); all solvents were bought from Merck–Sigma Aldrich (Darmastadt, Germany) (hydro fluoric acid (HF) is 48% in water, tetrahydrofuran (THF) and toluene are of anhydrous grade, ethanol is either ACS reagent or puriss p.a. grade). (3-Aminopropyl)triethoxysilane (APTES) was bought from Merck–Sigma Aldrich (Darmastadt, Germany) and used without further purification. BCAR was purchased from different vendors (Sigma-Aldrich, Alfa Aesar (Kandel, Germany) with purity >97%). Since this BCAR is extremely sensitive to oxidation mechanisms, we checked the reagent quality directly before all experiments by measuring its optical absorption spectrum to confirm the absence of detectable, already existing, oxidized species.

### 2.2. Synthesis of Porous Silicon

PSi was fabricated using an electrochemical cell made of Teflon. The silicon sample was mechanically clamped on a copper electrode. Since the electrochemical junction worked in reverse biased modes, neither additional doping nor metallic sputtering on silicon backside have been performed to improve Ohmic contact. The counter electrode was a platinum grid of about 6 cm of diameter assuring low polarizability of this electrode. The etching is performed under constant current conditions. We fabricate two types of PSi that differ in their porous structure: (i) Psi-A is etched from n-type 0.02 Ω cm silicon wafers using a solution of 16% *v*/*v* HF (48%) in ethanol, with an applied current density of 82 mA/cm^2^; (ii) Psi-B is etched from p-type 0.01 Ω cm silicon wafers using a solution of 6% *v*/*v* HF (48%) in water, with 0.5% of Triton-100 added as surfactant. Etching current density was 53 mA/cm^2^.

Most of the experiments have been done on PSi-A samples, while PSi-B has been used to check for any role of the pore structures on the release dynamics.

### 2.3. Relelase Studies of BCAR

The release of BCAR has been investigated also on PSi with different surface chemistry. We prepared two types of PSi surfaces: (i) the PSi_APTES_ was obtained by thermal oxidation of PSi-A on a hot plate at 200 °C in ambient atmosphere followed by silanizing it with aminopropyl-silane (1:100 = APTES:toluene) at 60 °C for 15 min; (ii) the PSi_H2O2_ was thermally oxidized as above and then placed in hydrogen peroxide (30% in water) for 24 h to further stimulate surface oxidation.

In all cases, all samples were thoroughly rinsed with ethanol before loading BCAR and gently dried under a nitrogen flow. PSi samples were loaded soon after their preparation to assure a hydrogen-terminated surface and, thus, a limited polarity. PSi samples were loaded by adding the required volumes of BCAR solution (THF has been used as solvent during the loading and the concentrations of the solutions are reported in Table 1) until the desired loading amount. The loading was done by repeated additions of drops of 10 to 20 μL, carefully dispersed onto the PSi area to achieve the best loading homogeneity, paying attention that the solution is confined within the porous area and avoiding coffee stain effects (Appendix A (ESI) reports the method used to check for the good dispersion of BCAR across the entire porous sample and to avoid coffee-stain surface agglomeration effects). No vacuum is applied during the loading as the capillary effect drives the infiltration of the loading solution within PSi pores. Release experiments were performed by placing the PSi sample in 3 mL of EtOH and using a magnetic stirrer for proper mixing (EtOH was used as it is a polar solvent in which BCAR is enough soluble to be detected by optical absorption measurements). All experiments were performed at ambient conditions.

### 2.4. Imaging Studies by Scanning Electron Microscopy (SEM)

SEM PSi cross sections were acquired soon after the samples preparation (to avoid surface oxidation and the need to coat the sample with conductive layer) on a Jeol JMS 7401F SEM (Tokyo, Japan) at either 5 kV or 10 kV of beam energy.

Auger (AES) measurements were performed at room temperature with a JEOL Ltd. JAMP-9500F (Tokyo, Japan) field emission scanning Auger microprobe system. AES line profiles were acquired with a primary beam of 10 keV. The take-off angle of the instrument was 0°. For Auger elemental analysis, an 8 nm probe diameter was used. Elemental images were acquired with a primary beam of 10 keV. The take-off angle of the instrument was 0°.

## 3. Results

### 3.1. PSi Characterisation

Scanning electron microscopy cross sectional images of PSi-A samples are reported in Figure 1.

From the analysis of the pore cross section of PSi-A, we estimate an average pore diameter of about 50 ± 10 nm. Pores of PSi-B samples are more branched, and it is difficult to estimate their average diameter from the cross section. A similar analysis performed from the SEM top view provide a value of 41 ± 7 nm (see Appendix A of the ESI for the details), still assuring a free diffusion of BCAR (see below).

PSi porosity has been estimated from interferometric optical measurements. Since the PSi samples are opaque, the interferometric analysis has been performed using a Fourier transformed infrared microscope. Two representative spectra are shown in Figure 2.

The reason to use FTIR spectroscopy is the opaqueness of PSi samples in the visible range that prevents the possibility to obtain the more precise analysis using visible frequencies.

By knowing the physical thickness of the layer *d*, the refractive index *n* is derived unambiguously using Equation (2):(2)$$d=\frac{N}{2W\sqrt{n^{2}-{\left(sin\theta \right)}^{2}}\;}$$
where *θ* is the incidence angle, *W* is the wavenumber interval considered and *N* is the number of periods contained within *W*. In case of normal incidence, as in our case, the above equation simplifies to Equation (3):(3)$$n_{eff}d∼\frac{N}{2W}$$

Once the effective refractive index is known, PSi porosity is estimated using the Bruggeman effective model (Equation (4)):(4)$$f\cdot \frac{n_{Si}^{2}-n_{eff}^{2}}{n_{Si}^{2}+2n_{eff}^{2}}+\left(1-f\right)\cdot \frac{n_{air}^{2}-n_{eff}^{2}}{n_{air}^{2}+2n_{eff}^{2}}=0$$
where *f* is the fraction of silicon in the porous layer while *n_Si_* and *n_air_* are the refractive indexes of silicon and air, respectively. The results are summarized in Table 2.

### 3.2. Auger Electron Spectroscopy Analysis

AES analysis performed on PSi cross-section (after loading of the sample with BCAR and before release of BCAR) reveals that the drug is homogeneously dispersed along the pore’s dept load and releases experiments h, as reported in Figure 3a,b. Sample surfaces after releasing BCAR show that carbon signal (Figure 3e) from BCAR is homogeneously colocalized with oxygen (Figure 3c), in the positions of BCAR crystallites showed by the SEM images (Figure 3f). 

### 3.3. Release Experiments

PSi-A samples were loaded with various amounts of BCAR using three solutions of different concentrations as summarized in Table 1.

The two cells labelled with N.P. (not performed) indicate experimental conditions that were too extreme to perform experiments: the highest concentrated one, because nucleation appears even during the loading; the lowest concentrated, because of difficulties in constraining the loading to the porous area through sequential addition of several liquid drops.

Releases never follow a Fickian profile (the choice of the diffusion models is described in detail in the Discussion section), and the only model that describes all the experimental data with small uncertainties on the fitted parameters is the 1st order kinetic one.

Figure 4a reports the release profiles normalized to the amount of loaded BCAR. Each curve is the average of three releases from different PSi samples and the shadowed regions are the standard deviations (SDs). By comparing samples loaded with the less concentrated solution Figure 4a1, we note that those loaded with the smaller amount of BCAR release the drug nearly quantitatively and with small SDs while, upon increasing the loaded amount of BCAR, PSi samples release less cargo and SDs increase.

On the contrary, by using the most concentrated loading solutions Figure 4a3 the release is always incomplete, irrespectively on the amount of BCAR loaded. Similarly, by increasing the concentration of the loading solution, while keeping the same amount of BCAR loaded (same symbols going from Figure 4a1–3 the normalized releases decrease and become less reproducible. Reflecting this, the SDs increased from about 5–6% on the more diluted samples up to 20% of the most concentrated ones. In all cases, the amount of BCAR loaded into each sample is from 23 (for most diluted samples) to 6 (for most concentrated samples) times below its solubility limit in the release medium volume. It is worth noting that incomplete releases happen for both highly loaded carriers as well as for lightly loaded using concentrated solutions. This observation indicates that BCAR aggregates in different forms within the nanopores depending on the details of the loading and that its actual form (molecularly dispersed, amorphous, crystalline, etc.) does not depend only on the amount of BCAR (e.g., its surface density), but rather on details such as how the solvent dries within the pores and the role of the liquid meniscus in re-distributing BCAR.

Considering that the two least loaded samples of Figure 4a1 release BCAR nearly quantitatively, we assume that no aggregates form under these loading conditions. In fact, since the BCAR absorption spectra shows no changes (see below), the non-released BCAR (~10%) is, probably, non-specifically adsorbed on PSi walls. The sample loaded with the maximum amount of BCAR (black squares) shows aggregates on the PSi surface from some minutes after the start of the release experiment. Similarly, Figure 4b reports the release from samples loaded with the 0.3 mg/mL solution: in this case, the release becomes non-quantitative at smaller loaded BCAR (blue stars). This fact correlates with the formation of macroscopic aggregates on PSi surface. Using 1.5 mg/mL aggregates are visible to the naked eye when only 3.8 µg of BCAR are loaded, while if 0.3 mg/mL is used, their formation is evident when at least 12 µg of BCAR is loaded. The presence of surface BCAR aggregates is confirmed by AES performed on the surface of PSi samples. Figure 3b shows an example of AES/SEM elemental maps confirming the presence of organic crystals on the surface of PSi samples in Figure 3c,e. These samples were loaded with 10.5 µg of BCAR and left in 3 mL of EtOH for 18 h. The formation of macroscopic organic crystals supports the fact that surface HN events are triggered by the release of the lipophilic molecules, while their presence after such long time confirms the poor solubility of this form of BCAR. Unfortunately, the limited amount of crystals formed on the PSi surface prevented us from obtaining detailed chemical and structural information about the crystals and to unveil the reason for their extremely slow solubility.

Since normalized releases do not account for the absolute flux of drug diffusing from the pores, the parameter of interest in case of drug delivery applications, Figure 4b reports the absolute release profiles (i.e., the relative ones multiplied by the total loaded BCAR). It is noteworthy that the absolute amount of released BCAR is not proportional to the loaded amount, but it passes through a maximum whose value depends on both the concentration of the loading solution and on the total amount of drug-loaded. For both the 0.17 mg/mL and 0.3 mg/mL solutions, the largest amount of BCAR is obtained for the intermediate loading (10.2 µg). On the other hand, the most concentrated loading solution produces significantly reduced releases independently from the loaded amount of drug.

To better clarify the experimental trends, we define a figure of merit (FOM) as the ratio between the loaded BCAR and the SD of the release:(5)$$\mathrm{FOM}=\frac{\left(\mu g\right)\;\mathrm{of}\;\mathrm{loaded}\;\mathrm{BCAR}}{\mathrm{SD}\;\mathrm{of}\;\mathrm{release}\;\mathrm{experiment}},$$

This FOM shows that the reproducibility of the release depends on the loaded amount of drug (with large FOMs indicating that BCAR is released nearly quantitatively and in a reproducible way).

Figure 5a reports how FOM varies across the datasets: when a 1.5 mg/mL loading solution is used, FOM always yields low values, while using the more diluted solutions, the FOM is low for a large amount of BCAR loaded, but it increases for smaller loaded quantities. A particular case is the sample loaded with 10.5 µg of BCAR using the 0.3 mg/mL solution: its FOM has an intermediate value (~8), the sample does not form macroscopic aggregates visible by naked eye, but it releases slightly less BCAR compared to the samples loaded with smaller BCAR amount (about 80% vs. 90%) and its SD is a bit larger (11% vs. 6%). These observations suggest that, when using a 0.3 mg/mL solution, the limiting amount of BCAR that can be loaded without macroaggregates formation is 10.5 µg.

Figure 5b shows how much BCAR is released versus the loaded amount. An optimal loading value exists to obtained quantitative and reproducible release of the drug. Considering the carrier properties, such optimal value of loading depends on its specific area as well as on its characteristic length scales. We tried to perform similar experiments on truly nanoporous samples (with pore diameter smaller than 20 nm) but BCAR always aggregates on the PSi surfaces during the loading (even using the most diluted conditions) and this prevents any possible release experiment.

To verify that no other experimental parameters may affect the releases, we check the dynamics of the release on both PSi-A and PSi-B samples, as well as on PSi-A with different surface functionalization. The results are reported in Figure 6.

All samples were loaded with 20 μL of 0.3 mg/mL BCAR solution, as this is the concentration at which a decrease in the amount of BCAR released starts (see Figure 4a2 blue stars).

Figure 6a compares the release from the two type of PSi we prepared, and it shows no differences neither on the dynamic nor in the amount of compound released. Figure 6b compares the release from PSi with different surface functionalization. Also, in this case, the releases are comparable within the SD and do not show significant variations. The inset of Figure 6b shows that the absorption spectra of released BCAR is not modified compared to the reference (black line), meaning that the drug is not appreciably oxidized from the loading and up to its release. These two analyses indicate that the fine details of pores structures and of surface chemistry do not play a significant role on the onset of HN events.

## 4. Discussion

In this article we demonstrate that HN appears during the release of poorly soluble molecule at the surface of nanostructured scaffolds even under highly diluted conditions. We use planar macroscopic PSi samples to highlight the role of the nanostructures present on the sample surface in decreasing the energy barrier for HN.

Our results indicate that to design effective DDS showing highly reproducible release kinetics, the maximum loading capacity of a drug carrier might not be a good parameter, if considered alone. Particularly for the case of highly lipophilic drugs, agglomeration of the loaded compound might happen at the carrier outer surface during the release of the drug and even under diluted conditions. Due to the stochastic nature of HN events, agglomeration produces a wide range of particle sizes, going from nano to micron-scale. Since the rates of dissolution of such different particles are markedly different, the variation in size also induces a large broadening of the release dynamics. Albeit we cannot demonstrate the formation of BCAR polymorphs (due to the the limited amount of agglomerated BCAR), the solubility of the solid particles formed at the PSi outer surface is an order of magnitude slower than the one of molecular BCAR, and they take up to tens of hours to completely dissolve, even under perfect sink conditions (BCAR loaded was from six to 23 times less concentrated that its solubility limit in release media).

The appearance of HN depends both on the total amount of drug loaded, as well as on the concentration of the drug into the loading solution. While the former is intuitively expected, the latter indicates that, albeit BCAR is macroscopically homogeneously distributed along the pore depth, the way it redistributes on the inner surface of the pores, during the subsequent additions of loading, creates different local environments that affect the release dynamics.

The observation of HN on sample outer surface, during the release, means that even if BCAR is solubilized (or solvated) within the pore, the energy landscape it experiences while reaching the carrier outer surface changes dramatically. This suggests that the nanostructures present on the outer surface have an increased surface energy with respect to those present inside the pore (possibly because of their smaller length-scale compares to the inner pore roughness), that easily induce HN.

Theoretically, the aptitude of BCAR to aggregate depends on the carrier surface chemistry, yet we obtained similar results from experiments done on PSi samples with different pore structure and surface functionalization (e.g., polar and non-polar). These measurements indicate that release dynamics are not heavily affected by the surface chemistry details (obviously the quantitative amount of HN might vary for different surface chemistry) and that, at least for non-polar molecules as BCAR, nucleation-aggregation phenomena are mainly driven by the high surface energy of the nanostructures and by the interfacial effects during solvent drying in the loading phase, while it is weakly related to the fine details of the carrier surface chemistry. Moreover, the line shape of BCAR absorption spectra suggests an excellent chemical inertness of the PSi carriers and their possible use to effectively protect easily oxidizable molecules.

The stochastic nature of HN greatly increases the variance of the release kinetic, degrading its reproducibility. Albeit this effect might not be fundamental for the ideal case considered here, where a nutraceutical integrator has been chosen as model molecule, the same effect will likely strongly limit the use of nanoporous DDS for other type of pharmaceutical components for which the release kinetics should be precisely known.

The increase of the variance in the release curves is proposed as a method to unveil HN events: compared to the standard ITM, the release curve provides an integral measure over the entire sample rather than a local analysis, the accuracy of which is determined by the spatial resolution of the microscopic technique used. In fact, ITM measures the time agglomerates take to form, often using optical methods, so its reproducibility is rather poor as it is determined by the stochastic nature of clusters formation [57,58].

On the contrary, our release-based method does not depend on a “spatial” resolution to detect the agglomerates. Instead, it directly measures actual concentration of molecular species. Thus:
release curves measure a global system response, averaged over the entire surface of the porous sample, rather than the local analysis given by the ITM whose precision depends on the spatial resolution of the technique used;the sensitivity achieved is much greater. For example, in our case agglomerates form and remain attached to the PSi inner surface, thus they will be invisible in standard ITM characterization based on optical scattering.


### About of the Choice of the Model to Fit the Data

In this section we discuss some of the many models proposed to fit release data and we describe the reason we chose the 1st order kinetic one. Fundamentals equations are Fick’s laws:(6)$$J=-D\cdot \;\nabla C\;\frac{\partial c}{\partial t}=D\cdot \nabla^{2}C$$

Obtaining an analytical solution to Equation (6) is a formidable task [59,60], particularly when dynamics changes of either the carrier structure or drug concentration must be considered. In these cases, the model has to account for moving interfaces, e.g., carrier swelling and/or erosion [61], to properly fit the releases. 

Being chemically inert, PSi shows no changes on its structure during the release. Thus, a release model should contain only parameters connected with the interactions between the drug and the pore surfaces. For example, when pore sizes are only a few times larger than the molecular species, then particular diffusion mechanism act [62]. Actually, for our PSi samples, the pore size is typically an order of magnitude larger than BCAR molecules, and molecular free diffusion is expected.

Recently, Jarosz et al. [63] proposed a composite model able to fit the initial burst and the following Fickian profile for the release of drugs from porous alumina. Since our releases do not show any burst, such a model cannot be applied in our experiments.

Moreover, considering that no substantial differences on the release dynamics have been outlined by different surface chemistry, we suppose that the drug is free to diffuse out of the pores without strong surface interactions (this assumption is demonstrated by the value of the diffusion coefficient found, see below). This is also supported by the non-polar nature of the BCAR that interacts with surfaces only via weak short-range van der Waals forces. Given these hypotheses, we considered only those models that describe monolithic systems and freely diffusing loaded molecules.

Equation (6) admits an analytical solution valid for drugs homogeneously dispersed in the matrix and with perfect sink conditions maintained during the release, in the form of a trigonometric series expansion [64] (Equation (7)):(7)$$\frac{M_{t}}{M_{\infty }}=1-\frac{8}{\pi^{2}}\overset{\infty }{\underset{n=0}{\sum }}\frac{1}{{\left(2n+1\right)}^{2}}exp\left(-\frac{D{\left(2n+1\right)}^{2}\pi^{2}t}{L^{2}}\right)$$
where *M_t_* and *M_∞_* are the cumulative release of drug at time *t* and at infinite time, *D* is the diffusion coefficient, and *L* is the thickness of the releasing layer.

Many simplified models to describe the release of drug homogeneously dispersed in thin slabs were also proposed. One of the most famous is the Higuchi model [65] that is valid when the drug concentration is much larger than its solubility in the matrix and it provides the well-known “square root” dependence:(8)$$M_{t}=A\sqrt{\left(2c_{0}-c\right)c_{s}Dt}$$
where *M_t_* is the amount of drug released at time *t*, *A* is the surface through which the release happens, while *c_0_* and *c_s_* are the initial drug concentration and the solubility of the drug in the carrier, respectively.

Another square root dependence can be derived from the analytical model (Equation (8)) if only the initial release time is considered (*M_t_*/*M_∞_* < 0.60). In this case the dynamics follows:(9)$$\frac{M_{t}}{M_{\infty }}=A\sqrt{\frac{Dt}{L^{2}}}$$

Since the square root dependence is obtained starting from models that assume completely different assumptions, it is generally regarded as an indicator for diffusion-controlled mechanisms (to note that same dependence is found also for the wetting of capillary by a liquid [66]).

More generally, and under certain assumptions [67], the analytical model can be reduced to a power law that is valid, again, only for the initial part of the release:(10)$$\frac{M_{t}}{M_{\infty }}=kt^{n}$$

In another case, the exponent has been used to classify the type of transport mechanism, considering the Weibull model [68]:(11)$$\frac{M_{t}}{M_{\infty }}=1-exp\left(-at^{b}\right)$$

It is worth to notice that a stretched exponential, as the Weibull model, is the limit of an infinite sum of exponentials (via Laplace transform):(12)$$\exp\left(-at^{b}\right)=\int_{0}^{\infty }P\left(\nu ,b\right)exp\left(-\left(\nu a\right)t\right)d\nu$$

Thus, this model should be used only when several time scales operates simultaneously creating the overall release.

Initially we fitted our experimental results using Equation (7) and we found that, most of the times, the fit procedure was robust and converges to find the correct experimental parameters (as the amount of BCAR loaded and the pore lengths). Yet, the confidence interval for both D and L were always huge (several orders of magnitude), indicating either a model weakness or a poor experimental set of data. None of the model described above were robust against our data. The only one that always fit all our experiments was the first order kinetic model. This model was proposed in [69] and it might be viewed as a Weibull model with a single exponential decay:(13)$$c\left(t\right)=A\cdot \left(1-e^{-kt}\right)$$
where *c(t)* is the concentration of the drug at time *t*, *k* is a proportionality constant, and *A* can be thought as the total amount of drug that can be released by the system. Since we are not interested in the details of the diffusion mechanism and considering that the structural details of all PSi samples are fixed (total area and pores depth), the value of the kinetic constant k is proportional to the flux of BCAR out-diffusing from the samples and is the global parameter we used to quantify and compare fluxes from different samples. To note that the fitting consider only the “fast” diffusion of molecular BCAR from the samples, while dissolution of aggregates (which is orders of magnitude slower as described below) is considered negligible, thus *t_∞_* is the time needed to molecular BCAR to exit from PSi pores. Table 3 reports the parameters of the fit using both Equation (7) (left grey part of the table) and the first kinetic model (Equation (13)).

Representative results for the fit using Equation (13), of two PSi-A samples of different thickness, are shown in Figure 7.

The fit (dotted line) follow the experimental data rather well and the releases show a significant variation in the BCAR diffusion. Intuitively, by increasing the layer thickness the saturation is retarded: it happens at about 50 min using the thinner (26 μm thick) PSi and at about 100 min with the thicker (46 μm thick) PSi. This fact was expected as the molecules diffuse for longer length on the thicker sample, but it also suggests that using this loading condition, BCAR molecules are adsorbed rather homogeneously along the whole PSi walls (without agglomerating at the surface), thus indirectly confirming the AES data. The fitting of the data with the 1st order model provides values of the diffusion coefficient of the same order of magnitude of those for liquid in liquid diffusion (10^−5^ cm^2^/s).

## 5. Conclusions

In this article, we demonstrate that the release of poorly soluble molecules from nanoporous materials is a complex process that might stimulate heterogeneous nucleation (HN) at carrier surface if the loaded amount of molecule is above a certain threshold. We characterize the release of BCAR from PSi samples of different surface polarities, and demonstrate that, for high BCAR loading concentrations, HN forms micro and macroparticles with markedly different solubility in a model solvent. The stochastic and slow dissolution rate of these particles degrades the reproducibility of the release experiments. The role of HN was demonstrated on a mesoscopic length scale, with Knudsen number falling in the transitional regime. Similar experiments done on macropores do not report such large variation of the release dynamics, while truly nanopores (diam. <20 nm) prevent the loading reasonable amount of molecule before forming surface crystals. This fact suggests that the energy surface landscape largely depends on the typical length-scale of the sample surface (in terms of roughness and pore diameters). Thus, even if the BCAR molecule is well solvated within the pore, as soon as it reaches the PSi surface, it might find a nucleation site that stimulates the growth of large solid particles even under dilute conditions.

We perform similar experiments using both polar and non-polar surfaces to rule out any significant differences due to surface interaction between the drug and the carrier. Thus, although we demonstrate this effect on a model system using a truly non-polar molecule, our results are general and demonstrate that, with poorly soluble compounds, there is a limit to the amount of drugs that can be loaded into a porous carrier while still obtaining reproducible release behavior.

Finally, we suggest the use of the release experiments to monitor the appearance of HN events on nanostructured materials, as it overcomes several limitations of the standard induction time method.

## Figures and Tables

**Figure 1 nanomaterials-10-01659-f001:**
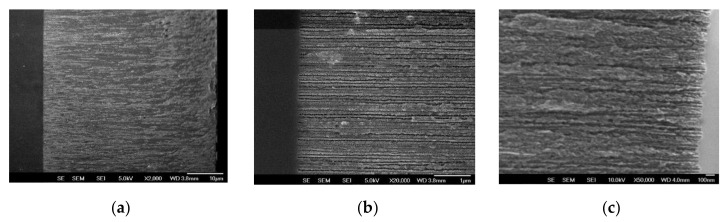
(**a**) Cross section of a PSi-A sample: porosity is homogeneous along the entire pore depth. (**b**) Higher magnification of the bottom part of PSi-A layer. (**c**) Cross section of PSi-B sample.

**Figure 2 nanomaterials-10-01659-f002:**
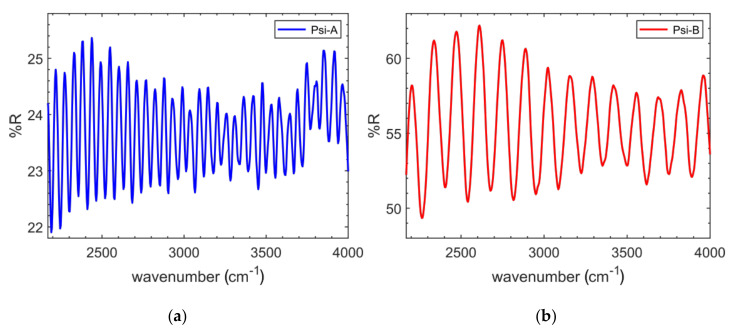
FTIR interferogram of (**a**) PSi-A sample and (**b**) of PSi-B sample.

**Figure 3 nanomaterials-10-01659-f003:**
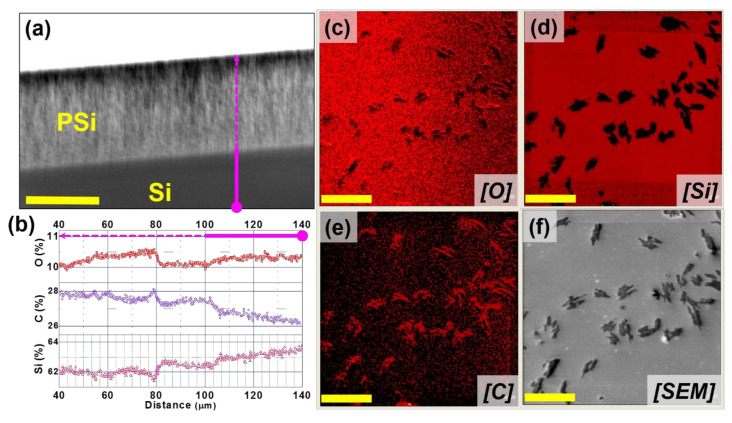
(**a**) The cross-sectional SEM image of a BCAR-loaded PSi-A sample and (in pink) the line used for the AES element profiles (before BCAR release). Scale bar: 30 mm. (**b**) AES elemental line profile along the pink line in the SEM image of this BCAR-loaded sample: red (O), purple (C), and brown (Si) of PSi. AES individual elemental mapping on the top side of PSi sample after releasing the BCAR: (**c**) oxygen, (**d**) silicon, and (**e**) carbon showing the presence of BCAR aggregates on the PSi surface (red and black color presents highest and lowest intensity of specific elements present respectively). (**f**) SEM image of top view of PSi. A scale bar of 60 µm applies to panels (**b**–**f**).

**Figure 4 nanomaterials-10-01659-f004:**
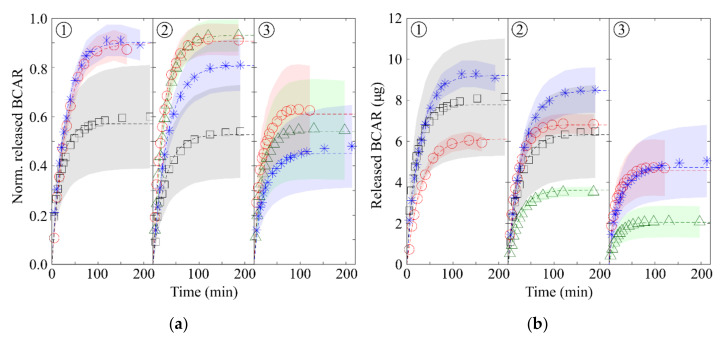
(**a**) Normalized BCAR release profiles from samples loaded with BCAR solutions of different concentrations. (**b**) Absolute release of BCAR from PSi samples. Concentration of loading solutions: (1) 0.17 mg/mL, (2) 0.3 mg/mL and (3) 1.5 mg/mL. Colors refer to the loaded BCAR amount: green triangles: ≈ 4 µg, red circles: ≈ 7 µg, blue asterisks: ≈ 10 µg and black squares: ≈ 12 µg.

**Figure 5 nanomaterials-10-01659-f005:**
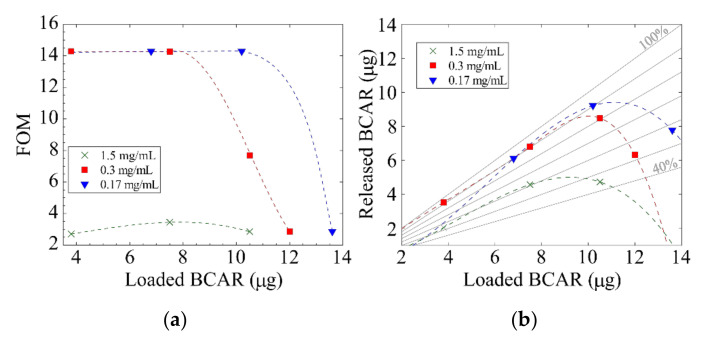
(**a**) FOM vs. the amount of loaded BCAR. Large FOM indicates nearly quantitative and reproducible release of the cargo. (**b**) Absolute release of BCAR vs. loaded amount for the different loading solutions. Gray oblique lines indicate percentage of release. Dotted lines are guides for the eyes.

**Figure 6 nanomaterials-10-01659-f006:**
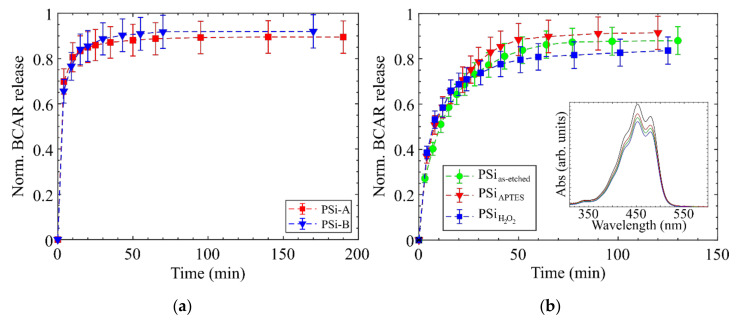
(**a**) Comparison between the two type of PSi structure investigated. Normalized BCAR release from (red squares) PSi-A and (blu triangles) PSi-B. (**b**) BCAR releases from PSi with different surface functionalization: (green circles) as-etched PSi-A; (red triangles) PSi-A functionalized with APTES; (blu squares) PSi-A treated with hydrogen peroxide. The inset shows the absorption spectra of BCAR, showing no lineshape modifications between the load and releases experiments (black curve reference of a BCAR solution). The different profiles between panel (**a**) and (**b**) are due to the different thickness of the two set of samples (23 μm in Figure 6a vs. 46 μm in Figure 6b). See Figure 7 for more details.

**Figure 7 nanomaterials-10-01659-f007:**
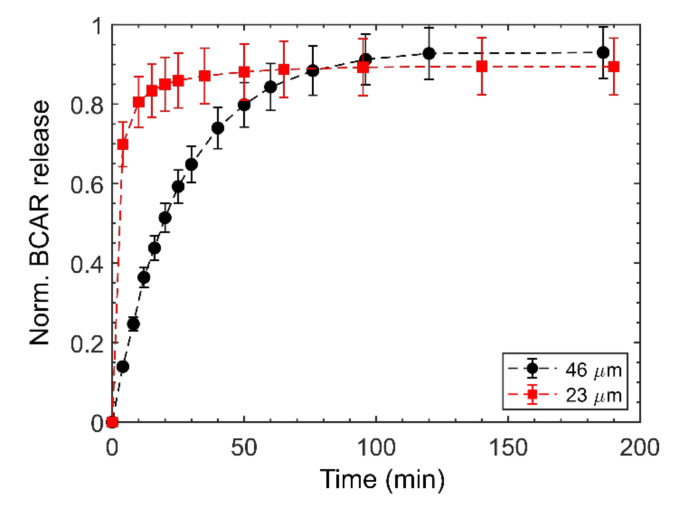
Experimental release curves fitted with the 1st order kinetic model for two PSi-A samples of different thickness. Error bars are the SD over three different releases from three different PSi samples.

**Table 1 nanomaterials-10-01659-t001:** BCAR loading solutions and loaded amounts.

BCAR Loading Sol. Conc. (mg/mL)	Loaded BCAR (g)			
1.5	3.8	7.5	10.5	N.P.
0.3	3.8	7.5	10.5	12.0
0.17	N.P.	6.8	20.2	13.6

N.P.—not performed

**Table 2 nanomaterials-10-01659-t002:** Summary of PSi parameters. *n_eff_* is the PSi effective index and *P* is the porosity.

PSi Type	Etching Recipe	Etch Rate (nm/s)	Thickness (μm)	*n_eff_*	*P* (%)
PSi-A	16% *v*/*v* HF (48%) in ethanol	52	46.5	2.2	45
PSi-B	6% *v*/*v* HF (48%) in water + 0.5%v Triton-100	22	26.5	2.0	55

**Table 3 nanomaterials-10-01659-t003:** Example of releases from both PSi-A and PSi-B samples fitted with the (left) analytical mode (Equation (7)) and (right) the 1st order kinetic (Equation (13)). *M_∞_* is the normalized release amount of BCAR, *L* is the pore length, *D* is the diffusion coefficient, *k* is the time constant.

	Analytical		1st Order Kinetic		
	PSi-A 46 μm	PSi-B 26 μm	PSi-A 46 μm	PSi-B 26 μm	
*M_∞_*	0.96	0.89	0.96	0.87	*M_∞_*
*L* (μm)	48	24	0.04	0.3	*k* (s^−1^)
*D* (cm^2^/s)	4 × 10^−6^	8 × 10^−6^	-	-	-
*R* ^2^	0.984	0.990	0.999	0.986	*R* ^2^

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
