# Peer review of "Surface Heterogeneous Nucleation-Mediated Release of Beta-Carotene from Porous Silicon"

_nanomaterials, 2020, doi:10.3390/nano10091659_

Round 1

Reviewer 1 Report

My comments are attached as a .pdf file.

Reviewer 2 Report

In the manuscript the authors discuss about a possible heterogeneous nucleation of the loaded cargo molecules on the support surface, after the release from silicon mesopores and before the dissolution process. Unfortunately, as figures 3e and 3f show, the presence of the BCAR crystals on the silicon surface before the release experiments rebuts the initial supposition of a heterogeneous nucleation. Instead, a crystal growth followed by redissolution phenomena could be possible. The authors should revise there experiments based on this facts and thus, the paper could not be accepted for publication in this form. If experiments will be repeated, the samples should present clean outer surface, otherwise the entire discussion should take into account the amount of the BCAR initially crystallized on the surface.

Some other aspects that should be clarify.
- Line 63 “Mesoporous length scales (from tens to few hundreds of nm) “ mesopores have diameters in the range 2 to 50 nm !
- Line 119 “the absence of detectable, already existing, oxidized species” needs to be rephrase
- Line 130 “the PSiAPTES was thermally oxidized”, change it with PSiAPTES was obtained from Psi-A by thermal oxidation (similar for PSiH2O2)
- If the loading area is not exactly the same for all the samples a specific loading values should be used (mass/area).
- DSC analysis on sacrificial samples could help for quantitative determination of loaded BCAR in crystalline phase (on the surface).
- Further details on the release experiments are necessary. Normally, in the release experiments the solution is stirred at constant rpm.
- Table 1. Etch(??) rate and P[%] are not defined. What are the differences between the thickness (the depth of the porous layer, d) and the pore length, L (Table 3) as the values are not the same (including fig. 7).
- It is not clear which Psi-A samples are represented in figures 3 and 7.

Reviewer 3 Report

The manuscript of Bettotti et al. is quite interesting and well-written. However, the authors have to address the following questions/remarks before it can be accepted for publication in Nanomaterials.

  1. Page 1, Introduction (line 31): It could be interesting if the authors give one example for each application domain cited for nanoporous materials.
  2. Page 1, Introduction (lines 34 to 40): It is quite strange to find such sentences at the beginning of an introduction. Usually, the introduction is devoted to a short state of the art and to situate the studied subject within the international scientific context. It is only at the end of the introduction that the authors research subject is introduced.

I suggest therefore to the authors to move these sentences at the end of the Introduction.

  1. Page 1, Introduction: The authors have to add references for nanoporous materials applications which are cited within the first paragraph.
  2. Page 6, Table 2: This table is not vey clear for me. May be I missed some information, but what do the numbers in the columns below “Loaded BCAR [µg]” represent? How were they obtained?
  3. Page 7, lines 235-236: Can the non-release BCAR be entrapped deeper in the Psi materiels? How can it be studied or evaluated?
  4. Page 7, line 250: How do the authors explain te very fast release of BCAR? Is it possible to obtain a nanomaterial showing a sustainable BCAR release?
  5. Page 10, line 363: Why do the authors add the title “4.1. About of the choice of the model to fit date” since there is no “4.2” or more paragraph in the Discussion part?
  6. Page 10, lines 375-377: I do not agree with this comment because when having a look to the profiles given in the “Results” part, the BCAR is released very rapidly within the first minutes of incubation. Such fast release corresponds to the burst effect which can be due either to a too high solubility of the drug into the incubation medium, or to very low (or bad) interactions between the drug and the carrier.

Round 2

Reviewer 2 Report

The authors modified the manuscript according to the  previous comments and could be considered for publication.

Just few supplementary observations:

  • preferable Etching rate instead of Etch rate
  • P is the porosity of the etched layer
  • Fig. 7 ...  PSi-A samples of different thickness, loaded with 20 μL of 0.3 mg/mL BCAR solution. (if this is the case, or whatever)

This manuscript is a resubmission of an earlier submission. The following is a list of the peer review reports and author responses from that submission.

Round 1

Reviewer 1 Report

This work by Piotto et al. focuses on understanding an important issue for development of drug carrier for poorly soluble drug molecules. Hetrogeneous nucleation causes a number of issues and often hinder clinical success of formulations. I recommend acceptance of this work after these corrections have been made.

  1. Page 2, Line 49: The mesopore size range is not defined as per IUPAC classifications, which defines pores between 2-50 nm as mesopores. Please correct.
  2. Page 2, Line 51: 'is' should be replaced by 'are'
  3. Methods part is very weak: a great deal of information is missing which migh affect the reporducibility of this work. For example, which power supply was used, what was the counter electrode, whch FTIR, the loading methods is very unclear. How many drops of what volume were added, was any vacuum applied to ensure complete infiltration of pores? The details provided in the SI are very poorly described, please elaborate the loading method in detail so other researchers can utilize the method in future for various applications.
  4. Table 1: it should be "etching recipe"
  5. AES results are not clearly described in the results section: Figure 3: what surface is it top view or cross-section? which colour represent which element in Fig c, d, and e. Figure 3 f is not mentioned in the caption.
  6. Figure 6: Assuming pSi as-etched is same as pSi-A, why is the release profile different between the two runs?
  7. Page 10, Line 344-345: I do not agree with the statement about no burst release. Figure 6 a shows a very large fraction of drug being released within the first few hours, which is a characteristic of burst release.
  8. Page 10, Line 346-349: This could be related to the size of the pores and less to the surface charge. Plz check IUPAC definition of the pore size and how molecules diffuse through different pore ranges. This could provide more insight.
  9. Most of the euqtions are not numbered
  10. Some of the important articles and book chapters in this field are not cited: Please include.
    1. Luminescent Porous Silicon Nanoparticles for Continuous Wave and Time-Gated Photoluminescence Imaging, Theranostics pp 185-198, Part of the Methods in Molecular Biology book series (MIMB, volume 2054), DOI https://doi.org/10.1007/978-1-4939-9769-5_13
    2. Self-Reporting Photoluminescent Porous Silicon Microparticles for Drug Delivery, ACS Appl. Mater. Interfaces 2018, 10, 4, 3200-3209.
    3. Porous silicon for drug delivery applications and theranostics: recent advances, critical review and perspectives, Expert Opinion on Drug Delivery, 14:12, 1407-1422.
    4. Porous Silicon Particles for Cancer Therapy and Bioimaging, Nanooncology pp 305-340, Part of the Nanomedicine and Nanotoxicology book series (NANOMED), DOI: https://doi.org/10.1007/978-3-319-89878-0_9

Reviewer 2 Report

Dear Authors, 

After reading your report "Surface Heterogeneous Nucleation-mediated Release of Beta-carotene from Porous Silicon Drug Delivery Systems" I am confronted to the feeling that the work, in spite of its undoubted experimental efforts, is only at an early stage, and requires further inputs before the conclusions you intend to extract can be confirmed. 

You can find below a list of the missing aspects this reviewer finds. You can find additional minor comments in the attached annotated version. 

  1. Abstract: Note that "macroscopic" may not be an appropriate word... is it visible to the eye? Note that nuclei may not be an appropriate word either. You may more specifically refer to nucleation site or cluster, depending on the context.
  2. Introduction: Note that approval for medical use is always specific to a system, not generic. So this shall require further details to i.e. the blood stream? gastric tubes?
  3. Introd.: Please revise bibliographic search according to the annotations in the attached annotated document. 
  4. Introd.: There are at least two missing aspects in this introduction:
    1) The diffusion for loading of the drug and its release is influenced by the solvent. It should be specified if the studies concern different solvents for both processes.
    2) The surface of the loaded nanoparticles can change back to hydrophobic (hydrophobic recovery) so that the release is not produced in the same surface conditions as for the drug load. 
  5. Methods: The loading-release experiments are not sufficiently explained. The solution properties? T? PSi is in the form of layer supported on Si? This is not realistic!!! Release in EtOH is not realistic!!!
  6. Results: The characterization of the porosity and pore size of the structures is poor!!! Look for details in attachment.
  7. Results: The characterization of loaded PSi interface is poor!!! Experiments should be presented at least at three moments. As prepared, after loading, after release. No differentiation of stages... no way to extract conclusions!!!
  8. The BCAR loading conditions table shows two evident missing conditions... Why?
  9. The main hypothesis falls into questioning when you describe samples without "macroaggregates" but readily exhibiting a decrease of release... Pg 7, line 247. 

In overall, this reviewer would advise a review of the experiments with PSi PARTICLES (open porosity) in realistic conditions (saline medium for release, 37ºC) and STEP BY STEP microscopic/Spectroscopic characterization. 
